**Subject Category:**
Biology (whole organism)

ecology/health and disease and epidemiology

plasmodium, house sparrow, *Passer domesticus*, decline, survival, population

**Authors for correspondence:**
Daria Dadam
e-mail: daria.dadam@bto.org
Andrew A. Cunningham
e-mail: a.cunningham@ioz.ac.uk

†Present address: British Trust for Ornithology, The Nunnery, Thetford IP24 2PU, UK.

# Avian malaria-mediated population decline of a widespread iconic bird species

Daria Dadam[1,†], Robert A. Robinson[2], Anabel Clements[1], Will J. Peach[3], Malcolm Bennett[4], J. Marcus Rowcliffe[1] and Andrew A. Cunningham[1]

[1]Institute of Zoology, Zoological Society of London, Regent's Park, London NW1 4RY, UK
[2]British Trust for Ornithology, The Nunnery, Thetford IP24 2PU, UK
[3]RSPB Centre for Conservation Science, Royal Society for the Protection of Birds, The Lodge, Sandy SG19 2DL, UK
[4]School of Veterinary Medicine and Science, University of Nottingham, Sutton Bonington Campus, Sutton Bonington LE12 5RD, UK

DD, 0000-0003-0466-5256; RAR, 0000-0003-0504-9906; MB, 0000-0002-5490-2910; JMR, 0000-0002-4286-6887; AAC, 0000-0002-3543-6504

Parasites have the capacity to affect animal populations by modifying host survival, and it is increasingly recognized that infectious disease can negatively impact biodiversity. Populations of the house sparrow (*Passer domesticus*) have declined in many European towns and cities, but the causes of these declines remain unclear. We investigated associations between parasite infection and house sparrow demography across suburban London where sparrow abundance has declined by 71% since 1995. *Plasmodium relictum* infection was found at higher prevalences (averaging 74%) in suburban London house sparrows than previously recorded in any wild bird population in Northern Europe. Survival rates of juvenile and adult sparrows and population growth rate were negatively related to *Plasmodium relictum* infection intensity. Other parasites were much less prevalent and exhibited no relationship with sparrow survival and no negative relationship with population growth. Low rates of co-infection suggested sparrows were not immunocompromised. Our findings indicate that *P. relictum* infection may be influencing house sparrow population dynamics in suburban areas. The demographic sensitivity of the house sparrow to *P. relictum* infection in London might reflect a recent increase in exposure to this parasite.

# 1. Introduction

Parasites have the capacity to affect animal populations by modifying host survival and/or reproductive success [1]. It is increasingly recognized that parasites influence biodiversity [2,3], and that infectious disease emergence poses a threat to wildlife conservation via 'spill-over' from domestic animals, but also increased contact between wildlife populations, as a consequence of factors such as habitat loss and fragmentation, range reduction and supplementary feeding [2,4,5]. Several parasites have been demonstrated as having demographic impacts on birds. *Plasmodium relictum* and its mosquito vector in Hawaii caused declines and extinctions among the bird community [6], *Mycoplasma gallisepticum* caused house finch (*Haemorhous mexicanus*) declines in eastern USA [7], West Nile Virus caused the declines of several North American bird species [8] and *Trichomonas gallinae* caused a marked decline of the greenfinch (*Chloris chloris*) population in Great Britain [9].

The house sparrow (*Passer domesticus*, hereafter sparrow) used to be one of the most abundant species of birds in the UK, but its population has declined markedly, including in urban/suburban areas [10]. Populations have also declined throughout much of Western Europe [11], in India [12] and in North America [13]. Several hypotheses have been proposed and tested to explain the decline of the sparrow in the urban/suburban environment, but none to date has received strong and consistent support. Bell *et al.* [14] found a negative correlation between counts of sparrows in gardens and Eurasian sparrowhawk (*Accipiter nisus*) activity. By contrast, Peach *et al.* [15] found a positive relationship between adult sparrow abundance and Eurasian sparrowhawk activity but no relationship between temporal trends in sparrow abundance and hawk activity. Studies on the impact of predation by domestic cats (*Felis catus*) have reached no definitive conclusions. Sub-lethal effects of cat predation on passerines, including the sparrow, have been suggested, and a modelling approach has pointed to a possible effect on reproductive success [16]. However, breeding success is not a factor in the sparrow decline in the UK [17,18], whereas decreased overwinter survival [19], particularly of juveniles [20], has been identified as driving this population decline. A study in Leicester (UK) found a lower fledging rate when the diet of nestlings contained a high percentage of vegetable matter compared to invertebrates [17]. These results informed a large-scale supplementary feeding experiment in London, which showed that, while mealworm supplementation increased fledging success, it did not enhance recruitment of more juveniles into the breeding population [18]. Rather, as before, overwinter survival was identified as the likely demographic constraint on population growth [18]. Low overwinter survival was found to be unrelated to food availability by a year-round seed supplementation experiment in London [15]. The onset of the UK sparrow decline coincided with an increase in small particulate pollution from diesel engines, especially in cities, thus this has also been hypothesized as a causal factor [21], possibly by causing immunosuppression [22]. Thus, although several hypotheses have been proposed to explain sparrow population declines in the urban/suburban environment, no conclusive cause has yet been identified. As far as we are aware, parasitism has not been previously investigated as a potential cause of the declines; indeed, parasites are generally understudied as a potential cause of avian population declines [23].

Here, we investigate possible associations between parasitism and sparrow demography by relating parasite prevalence and intensity of infection to host survival and local population change using sparrows in London, UK, where sparrow abundance has declined by 71% since 1995 [24].

# 2. Material and methods

## 2.1. Study sites

The study was conducted between November 2006 and September 2009 at 11 sites across London (figure 1) as part of a wider investigation into house sparrow population declines [18,19]. Each study site was centred on an aggregation of territorial pairs (a colony) where most of the subsequent trapping and re-sightings took place (below). All sites were in residential suburban areas and at least 4 km apart in order to ensure demographic separation; natal and breeding dispersal average less than 0.3 km in sparrows and rarely exceed 2 km [25–27].

## 2.2. Measuring temporal changes in sparrow abundance

Annual surveys of male sparrows exhibiting territorial behaviour (chirping) were used to measure changes in local breeding populations between 2005 and 2009. Surveys involved fixed transects which

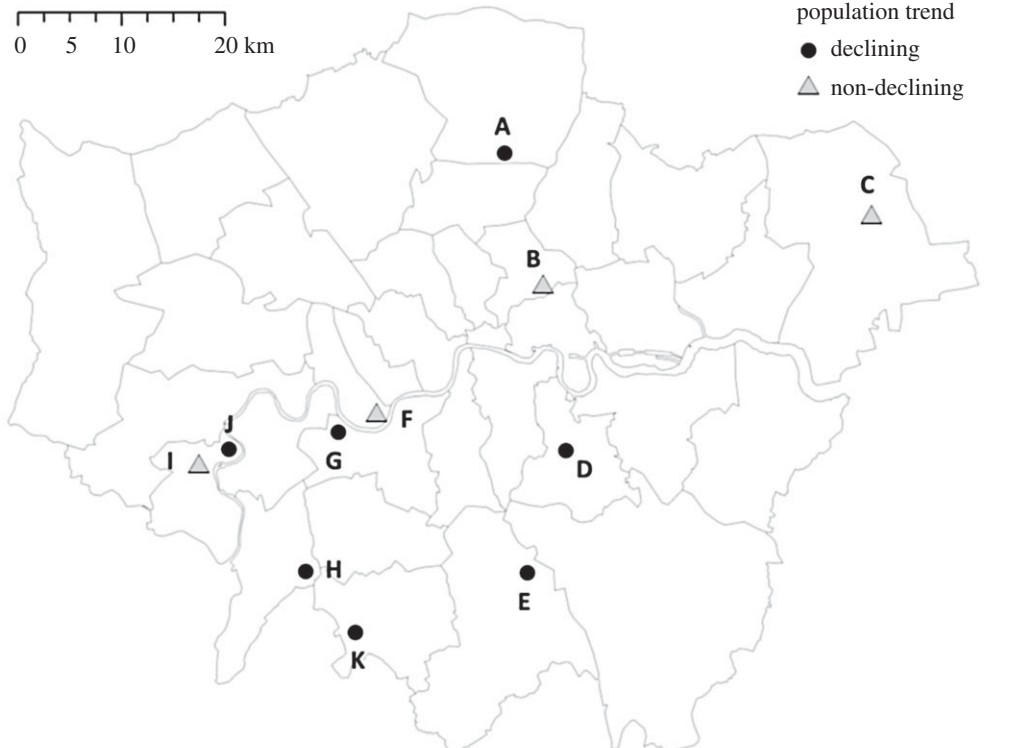

**Figure 1.** Map of house sparrow (*Passer domesticus*) study sites in London, indicating whether or not counts of territorial male house sparrows showed a declining population trend over the period 2005 – 2009 (Background map: [18]).

were walked twice each year by trained observers on dry, still mornings between mid-March and mid-May (with one month intervals between surveys), to count all male sparrows, distinguishing chirping (i.e. displaying breeding activity) from non-chirping individuals [18]. Each survey covered a central core area centred on a colony (mean core area = 1.7 ha, range 1–2.6 ha) plus a surrounding buffer area which extended 200 m beyond the core area (mean core plus buffer = 24.5 ha, range 19.1–36.5 ha) [18].

The highest of the two counts of chirping males obtained in each year within the core plus buffer area was taken as a measure of local breeding population size at each site [18] (electronic supplementary material, table S1). For each study site, we modelled (logarithm) annual male count as a function of year using a generalized linear model (GLM) and assuming a Poisson error distribution. The linear year effect provided an estimate of the average annual rate of change in abundance of territorial male sparrows (or local population growth rate) at each site over the 5 years study period.

## 2.3. Bird capturing and resighting

Sparrows were caught in mist-nets at each of the 11 study sites and sampled for parasites year-round for 3 years (from November 2006 until September 2009). Most sparrows were caught close to seed feeders located in core survey areas. Sites were selected to have similar suburban infrastructure and other habitat characteristics and to be suitable for the regular monitoring, catching and sampling of birds. Sites were visited at least once every three months (once per season) if a minimum of 10 individual sparrows had been sampled, or more often to achieve this minimum sample size within each season. In practice, multiple visits per site per season were carried out at most sites. Sparrows were marked with a uniquely numbered metal ring and three colours of plastic ring (two rings on each leg in a unique combination), aged (adults or juveniles) and sexed when possible [28]. On capture, each individual sparrow was placed in a clean cloth bag, which was used only once during each capture session and was washed at 90°C with detergent between sessions to kill and remove any parasites therein.

At five study sites (B, F, H, I and K, figure 1) a systematic mark-resighting programme took place from September 2007, following a pilot study in the winter 2006/2007 involving one site (H), in order

to measure overwinter survival [15]. Sparrows were individually colour-ringed (one metal ring plus three plastic colour-rings, two on each leg) and then resighted at four-week intervals between September and March (six to seven surveys per site during winters 2006/2007 (pilot), 2007/2008 and 2008/2009) by trained fieldworkers walking a fixed transect route covering all accessible parts of each study area [15].

## 2.4. Gastro-enteric parasites

Faeces were collected directly from captured birds by swabbing the cloaca using a sterile swab with charcoal medium (Transwab for aerobes and anaerobes, Medical Wire Equipment Co. Ltd, Corsham, UK), or from the bird bag as a faecal sac using 1.5 ml microcentrifuge tubes (Microcentrifuge with screw caps, Neptune, VWR, Pennsylvania). If the dropping was too small for collection, it was sampled using a sterile swab. Faecal samples and swabs were stored refrigerated at 4°C and processed within 24 h. Due to the small amounts of faeces collected from each bird, we used the faecal droplet technique to identify the relative abundance of coccidian (scored from 0 to 5) [29] and helminth parasites in each faecal sample (but not swab, which was used only for bacteriology) [30,31], counting all observable helminth parasites and ova.

Each faecal sample and swab were cultured for bacteria using all of the following: xylose-lysine deoxycholate (XLD) agar (QCM Laboratories, London, UK) and Colombia blood agar supplemented with 5% sheep blood (QCM Laboratories or E&O Laboratories, Burnhouse, Scotland) under aerobic conditions for 48 h at 37°C; Campylobacter blood-free selective medium (QCM laboratories; modified CCDA-Preston) under microaerophilic conditions at 37°C; selenite Salmonella-enrichment broth (QCM Laboratories or E&O Laboratories) under aerobic conditions for 24 h at 37°C followed by subculture on XLD agar, also under aerobic conditions. Bacterial isolates were identified using colony morphology, Gram's staining and biochemical properties using analytical profile index (API) 20 Enterobacteriaceae biochemical test strips (API-BioMerieux, Marcy l'Etoile, France). Suspected isolates of Salmonella spp. were submitted to the Laboratory of Enteric Pathogens, Public Health England for confirmatory testing.

## 2.5. Haemoparasites

Each captured sparrow was blood-sampled from the brachial vein after cleaning the overlying skin using a mono-use pre-injection swab (Sterets Pre-injection swab, Medlock Medical, Oldham, UK). The vein was pricked with a mono-use needle (Becton Dickinson Microlance, BD, New Jersey) and the initial drop of blood was collected onto a microscope coverslip (VWR Int.) and used to make a blood smear on a pre-washed, pre-polished glass slide (Menzel-Glaser, Brunswick, Germany). Blood smears were air-dried and transported to the laboratory where they were fixed using 100% methanol on the day they were made and were subsequently stained with May–Grünwald–Giemsa within a week of sampling. Stained smears were examined microscopically for haemoparasites by screening 100 fields of view at 1000× magnification at the centre of smear to avoid biases in counts, as there may be differential distribution of haemoparasites between the edges and the centre of blood smears [32]. Parasite numbers (i.e. intensity of infection) were expressed as the number of each parasite species per 10 000 erythrocytes for Plasmodium spp. and per 100 leucocytes for Atoxoplasma spp.

Additional blood was collected using a heparinized capillary tube (Spirocap 100 mm, Bilbate Ltd, Hastings, England) which was sealed with Cristaseal (Hawksley, Lancing, UK). Following sampling, a cotton ball was applied firmly to the site of venepuncture until haemostasis was achieved. All parasites seen on microscopic examination of the blood smears were identified according to [32,33], and despite recent taxonomic developments placing Atoxoplasma and Isospora as synonyms [34], we identified with the term 'Atoxoplasma' the extraintestinal asexual phase of some Isospora spp. that are found in monocytes and lymphocytes in the blood [32]. Isospora spp., therefore, identified the intestinal phase of the parasite, while Atoxoplasma spp. identified the phase found in the blood.

DNA was extracted using the standard protocol in the Qiagen DNAEasy blood and tissue extraction kit [35] from pelleted blood cells from 29 sparrows, with each study site being represented by samples from two or three birds that had tested positive for Plasmodium spp. on examination of blood smears. These samples were then examined for haemoparasites using a nested polymerase chain reaction (PCR) which amplifies a 478 bp fragment of mitochondrial cytochrome b gene shared by Plasmodium spp. and Haemoproteus spp. [36]. Positive and negative controls were included in each PCR run. PCR products (6−8 μl) were run on 2% agarose gels stained with ethidium bromide and visualized under UV light. A sample that showed a band of the expected size (450−600 bp) was considered positive

[36]. Amplicons from the 29 individuals were submitted for sequencing commercially (Beckman Coulter Genomics, Essex, UK). Sequences were edited and aligned using Mega 5.1. [37] and compared to their closest match in GenBank using NCBI nucleotide BLAST search. Also, sequence data were compared to those of other haemoparasites in the MalAvi database [38].

## 2.6. Relationships between parasite infection and local sparrow population growth

*Plasmodium relictum*, *Atoxoplasma* spp. and *Isospora* spp. (hereafter *Plasmodium*, *Atoxoplasma* and *Isospora*, respectively) were the only parasites found for which prevalences were high enough to explore potential relationships between the degree of parasitism and host demography. GLMs were used to test for relationships between local sparrow population growth rates and parasite prevalence (proportion of infected individuals at each site) and intensity (mean number of each parasite in infected individuals). Abundance of sparrows at each site in each year was modelled as a function of year (continuous variable), site (11-level factor) and the interaction between year and either mean infection prevalence or intensity. Observations relating to each individual colony were weighted within the GLM by the number of birds tested (for prevalence) or the number of positive individuals detected (for intensity) [39]. All models incorporated a quasi-Poisson error structure which corrects for overdispersion by multiplying the estimator by a variance term (or scale factor [39]). Separate analyses were conducted first based on infection rates of all birds, and then of adults and juveniles separately. For the latter, site was a nine-level factor because no juveniles were sampled at two sites; this prevented us using an information-theoretic approach for model selection. Too few individuals were sexed to allow sex-specific analyses.

## 2.7. Effect of parasite infection intensity on individual host survival

We used Cormack–Jolly–Seber mark–recapture models (in package RMark [40]) to test whether the monthly overwinter survival of individual sparrows was related to the intensity of *Plasmodium* or *Atoxoplasma* infection at capture (specified as an individual-level covariate) [41] using birds of known infection status and intensity. The number of infected individuals was too small to test for survival in relation to *Isospora*. The starting cohort was those colour-ringed individuals either captured or resighted during September. Only those individuals with at least one capture or sighting during the winter period (October to February), after any post-breeding dispersal had occurred, were included in the analyses ($n = 118$ for *Plasmodium* and $n = 117$ for *Atoxoplasma*). The analysis proceeded in two stages: first we used an information-theoretic approach to assess whether survival and resighting probability were affected by the factors: bird age, site and year. We then tested for the effects of parasite infection status on individual survival in models that allowed for any important age, site or year effects. By adopting this two-stage approach, we maximized our statistical power to detect effects of parasite status on survival. The three most parsimonious models (together accounting for more than half of the AIC weight) included a constant resighting rate with additive age, year and site effects on survival (electronic supplementary material, table S2). Subsequent tests for effects of parasite status on survival therefore included additive age, year and site effects on survival, and a constant resighting rate. We could not test for any effect of sex on survival or resighting probability as we could not reliably distinguish between juvenile males and females.

## 2.8. Impact of $NO_2$ pollution on house sparrow population growth, parasite prevalence and infection intensity

Evidence is accumulating that air pollutants derived from the combustion of fossil fuels can have deleterious impacts on birds, including respiratory distress, elevated stress, immunosuppression and impaired reproductive success [42], and a recent study in London showed that spatial variation in sparrow abundance is negatively correlated with nitrogen dioxide levels [15]. We therefore used GLMs to test for relationships between local $NO_2$ concentrations and (i) sparrow population growth rates and (ii) parasite prevalence and intensity measures of *Plasmodium*, *Atoxoplasma* and *Isospora*. Ground level $NO_2$ concentrations were derived from a kernel-based pollution dispersion model (King College's Air Pollution Toolkit) which combines point measures, traffic flows and emissions with hourly meteorological data to predict air pollution levels at a 20 m-grid resolution across London [43]. Predicted $NO_2$ levels from 2004 were averaged across the whole year, and then matched spatially to

**Table 1.** Prevalence of haemoparasites (based on microscopical examination of blood smears) and of gastro-enteric parasites (based on microscopical examination of faecal samples) in house sparrows according to age class.

| parasite | adult prevalence % (number of positive sparrows/number examined) | juvenile prevalence % (number of positive sparrows/number examined) | total prevalence % (number of positive sparrows/number examined) |
| --- | --- | --- | --- |
| *Plasmodium relictum* | 68% (126/185) | 80% (156/195) | 74% (282/380) |
| *Atoxoplasma* sp. | 28% (51/182) | 32% (62/194) | 30% (113/376) |
| *Leucocytozoon* spp. | <1% (1/185) | 0% (0/193) | <1% (1/378) |
| microfilaria | 0% (0/183) | <1% (1/195) | <1% (1/378) |
| *Isospora* spp. | 27% (28/103) | 35% (48/139) | 31% (76/242) |
| Cestode eggs | 8% (8/103) | <1% (1/139) | 4% (9/242) |

the study colonies. A similar GLM model to that described in §2.6 was used to test whether local $NO_2$ levels were related to population growth rate, with $NO_2$ replacing the parasite infection term. The GLM incorporated a quasi-Poisson error structure to correct for overdispersion (see above for more details). Binomial and Gamma error structures were incorporated into GLMs testing for relationships between $NO_2$ levels and parasite prevalence and intensity, respectively. Analyses were weighted by the number of birds tested (for prevalence) or the number of positive individuals detected (for intensity; see above). All data analyses were undertaken using R v. 3.2.1 [44].

# 3. Results

## 3.1. Gastro-enteric parasites

Of bacteriological examinations performed on faecal samples from 271 sparrows, one was positive for *Salmonella* Typhimurium DT56; no other pathogenic bacteria were isolated. The most prevalent eukaryotic gastro-intestinal parasites were coccidia, with 31% of 242 sparrows testing positive for coccidian oocysts, all morphologically characteristic of the genus *Isospora* (table 1). Mean intensity of infection varied between mean scores of 0–3.4 across sites (table 2). Fewer than 4% of samples tested positive for cestode eggs (table 1).

## 3.2. Haemoparasites

All 29 sequences obtained from PCR-positive samples had 100% identity with one of two strains of *P. relictum*: SGS1 (GenBank ref. JX196867, $n = 24$) and GRW11 (GenBank ref. JQ778277, $n = 5$). No other haemosporidians were detected (table 1).

The overall prevalence of *Plasmodium* infection, based on microscopic examination of blood smears, averaged 74% ($n = 380$, table 1), and varied between 50 and 100% across sites (table 2). Mean intensity of infection varied from 5.5 to 27.9 parasites per 10 000 erythrocytes. Overall, *Atoxoplasma* prevalence was 30% and varied from 0 to 50% across sites (table 2). Mean intensity of *Atoxoplasma* infection ranged from 1 to 4.4 parasites per 100 leucocytes (table 2).

Samples of blood and faeces were available from the same individual for 192 birds. *Isospora* and *Atoxoplasma* co-infection occurred in only one bird while 115 (60%) birds showed no infection with either parasite. Thirteen individuals (7%) were positive for *Isospora* but not *Atoxoplasma*. Sixty-two birds (32%) were positive for *Atoxoplasma* but not *Isospora*. Co-infection with *Isospora* and *Plasmodium* occurred in 10 (5%) birds. Eighty-four of 374 individuals (22%) were positive for both *Plasmodium* and *Atoxoplasma*. No bird was positive for all three of these parasites (electronic supplementary material, table S3).

## 3.3. Relationship between parasites and house sparrow population growth

Population growth was negative at seven of the study sites and positive at four (figure 1). The relationship between parasites and population growth varied with parasite species, infection prevalence and infection intensity (figure 2 and table 3). Population growth was unrelated to

**Table 2.** Prevalence (number infected/number sampled) and mean intensity (calculated as number of parasites detected per individual sample) of *Plasmodium relictum*, *Atoxoplasma* sp. and *Isospora* sp. Parasite intensity of *P. relictum and Atoxoplasma* sp. is expressed as the number of parasites per 10 000 erythrocytes, while intensity of *Isospora* sp. infection is a mean scoring system (standard deviation in brackets; see Material and methods for details). Site trend shows the average annual growth rate (λ) (expressed on a natural logarithm scale).

| site name | site code | site trend | Plasmodium relictum | | Atoxoplasma sp. | | Isospora sp. | |
| | | | prevalence (%) | intensity | prevalence (%) | intensity | prevalence (%) | intensity |
|---|---|---|---|---|---|---|---|---|
| Enfield | A | −0.116 | 85.7 (12/14) | 5.85 (± 1.2) | 20.0 (3/15) | 1.0 (±0) | 0 (0/6) | n.a. |
| Walthamstow | B | 0.171 | 78.6 (22/28) | 10.0 (±7.6) | 39.3 (11/28) | 2.1 (±0.38) | 11.7 (2/17) | 2.0 (±0.29) |
| Romford | C | 0.063 | 62.5 (5/8) | 9.12 (±4.2) | 37.5 (3/8) | 1.7 (±0.28) | 0 (0/4) | n.a. |
| Camberwell | D | −0.015 | 50 (3/6) | 22.33 (±10.3) | 50.0 (3/6) | 1.0 (±0) | 0 (0/7) | n.a. |
| Chessington | E | −0.073 | 58.1 (18/31) | 5.48 (± 3.3) | 16.1 (5/31) | 1.4 (±0.2) | 0 (0/15) | n.a. |
| Fulham | F | 0.162 | 80.3 (45/56) | 6.78 (± 2.2) | 37.5 (21/56) | 2.2 (±0.27) | 0 (0/33) | n.a. |
| Putney | G | −0.067 | 50 (14/28) | 19.46 (±27.9) | 39.3 (11/28) | 2.6 (±0.83) | 0 (0/15) | n.a. |
| New Malden | H | −0.224 | 83.1 (84/101) | 27.89 (±4) | 11.3 (11/97) | 1.8 (±0.2) | 17.6 (12/68) | 3.2 (±0.16) |
| Uxbridge | I | 0.125 | 72.3 (47/65) | 7.27 (±2.1) | 46.2 (30/65) | 2.8 (±0.34) | 6.67 (3/45) | 3.3 (±0.35) |
| Leyton | J | −0.177 | 100 (2/2) | 16.0 (±5) | 0 (0/1) | n.a. | 0 (0/1) | n.a. |
| Sutton | K | −0.043 | 73.1 (30/41) | 7.36 (± 2.05) | 36.6 (15/41) | 4.4 (±0.77) | 29.0 (9/31) | 3.4 (±0.2) |

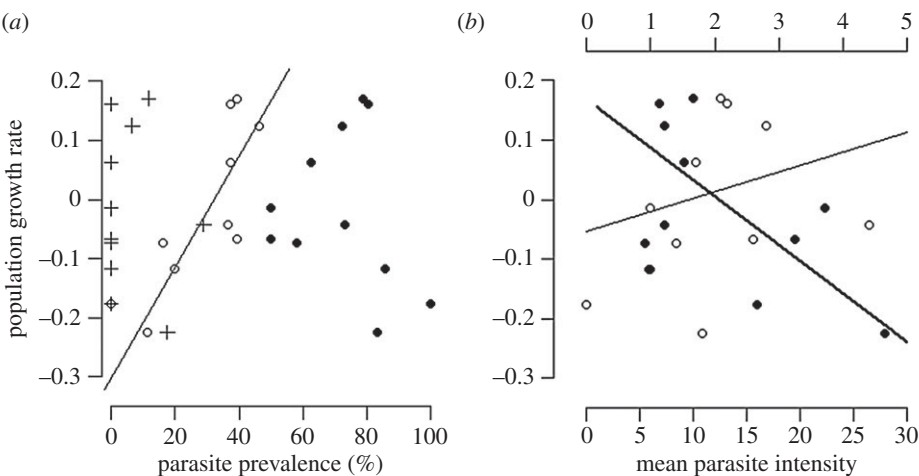

**Figure 2.** Prevalence (*a*) and mean intensity (*b*) of *Plasmodium relictum* (solid circles), *Atoxoplasma* sp. (open circles) and *Isospora* sp. (crosses) in house sparrow (*Passer domesticus*) adult and juvenile age classes combined in relation to observed 5-year population growth rate. Each point with the same symbol represents a distinct house sparrow population. Trend lines for *Atoxoplasma* (thin line) also shown. In (*b*), top *x*-axis refers to *Atoxoplasma* and *Isospora*. Trend lines for *Plasmodium relictum* (thick line) and *Atoxoplasma* (thin line) also presented.

**Table 3.** Summary of associations between prevalence and intensity of *Plasmodium, Atoxoplasma* and *Isospora*, year and population size across 11 sites. Estimates $\pm$ standard error (s.e.) of the interaction between year and parasite intensity or prevalence are reported, alongside the *F*-test and its significance. Prevalence of parasite infection is calculated as number infected/number sampled, while intensity of infection is calculated as number of parasites detected per individual. Significant results at 95% level in italics.

| parasite | measure | age | interaction estimates $\pm$ s.e | F | p-value |
|---|---|---|---|---|---|
| plasmodium | prevalence | both | $-0.489 \pm 0.362$ | 1.836 | 0.183 |
| plasmodium | prevalence | adults | $-0.137 \pm 0.234$ | 0.345 | 0.56 |
| plasmodium | prevalence | juveniles | $-0.312 \pm 0.337$ | 0.862 | 0.359 |
| plasmodium | intensity | both | $-0.012 \pm 0.003$ | 13.324 | *<0.01* |
| plasmodium | intensity | adults | $-0.003 \pm 0.013$ | 0.054 | 0.817 |
| plasmodium | intensity | juveniles | $-0.014 \pm 0.004$ | 13.906 | *<0.001* |
| atoxoplasma | prevalence | both | $0.908 \pm 0.209$ | *19.104* | *<0.001* |
| atoxoplasma | prevalence | adults | $0.731 \pm 0.179$ | *16.845* | *<0.001* |
| atoxoplasma | prevalence | juveniles | $0.776 \pm 0.232$ | *11.339* | *<0.01* |
| atoxoplasma | intensity | both | $0.127 \pm 0.061$ | *4.273* | *0.046* |
| atoxoplasma | intensity | adults | $0.535 \pm 0.173$ | *9.706* | *<0.01* |
| atoxoplasma | intensity | juveniles | $0.065 \pm 0.088$ | 0.5383 | 0.468 |
| isospora | prevalence | both | $-0.612 \pm 0.376$ | 2.659 | 0.111 |
| isospora | prevalence | adults | $-0.014 \pm 0.183$ | 0.006 | 0.938 |
| isospora | prevalence | juveniles | $-0.9130 \pm 0.421$ | *4.742* | *0.037* |
| isospora | intensity | both | $-0.087 \pm 0.289$ | 0.089 | 0.769 |
| isospora | intensity | adults | n.a. | n.a. | n.a. |
| isospora | intensity | juveniles | $-1.439 \pm 0.678$ | 4.575 | 0.058 |

*Plasmodium* prevalence (for both age classes: $F_{1,42} = 1.836$, $p = 0.183$, $n = 380$; for adults: $F_{1,42} = 0.345$, $p = 0.56$, $n = 185$; for juveniles: $F_{1,34} = 0.862$, $p = 0.359$, $n = 195$). However, population growth was negatively associated with the mean intensity of *Plasmodium* infection (for both age classes: $F_{1,42} = 13.3$, $p < 0.01$, $n = 282$ (figure 2*b*); for juveniles: $F_{1,34} = 13.9$, $p < 0.001$, $n = 156$), but not for adults

**Table 4.** Testing for linear relationships between individual house sparrow survival and the intensity of *Plasmodium relictum* infection. The table presents candidate models of mark-resighting data parametrized in terms of apparent survival (phi) and resighting probability (*p*) and ranked by AICc. Models allow for effects of sparrow age, year and site, as well as *Plasmodium* intensity (Plasm). The table includes the model definition, the number of parameters in the model (K), the difference between the AICc of the model in question and the best model (delta AICc), the AICc weight and the (age-specific) parameter estimates (logit scale Beta $\pm$ s.e.) for any relationships involving Plasmodium.

| model | K | $\Delta$AICc | AICc weights | age | beta |
|---|---|---|---|---|---|
| Phi(Plasm)p(.) | 3 | 0.000 | 0.262 | n.a. | $-0.012 \pm 0.005$[a] |
| Phi(age $\times$ Plasm)p(.) | 4 | 0.424 | 0.212 | juvenile | $-0.011 \pm 0.005$[a] |
|  | 4 | 0.424 | 0.212 | adult | $-0.027 \pm 0.013$[a] |
| Phi(age $+$ Plasm)p(.) | 4 | 0.895 | 0.167 | both | $-0.013 \pm 0.005$[a] |
| Phi(age $+$ year $+$ Plasm)p(.) | 5 | 2.660 | 0.069 | both | $-0.014 \pm 0.005$[a] |
| Phi(.)p(.) | 2 | 3.036 | 0.057 | both | n.a. |
| Phi(age $+$ year $+$ Plasm $+$ site)p(.) | 9 | 3.538 | 0.045 | both | $-0.019 \pm 0.007$[a] |
| Phi(age $+$ Plasm $+$ site)p(.) | 8 | 3.915 | 0.037 | both | $-0.017 \pm 0.007$[a] |

[a]Significant result based on confidence intervals. Plasm = *Plasmodium relictum*. K = number of parameters; Phi = survival; p = recapture rate.

alone ($F_{1,42} = 0.05$, $p = 0.82$, table 3, $n = 126$). Population growth was positively related to the prevalence (for both age classes: $F_{1,42} = 19.1$, $p < 0.001$, $n = 376$ (figure 2a); for adults: $F_{1,42} = 16.8$, $p < 0.001$, $n = 182$; and for juveniles: $F_{1,34} = 11.3$, $p < 0.01$, $n = 194$) and intensity (for both age classes: $F_{1,42} = 4.273$, $p = 0.045$, $n = 113$) of *Atoxoplasma* infection (figure 2b), the latter being significant for adults ($F_{1,38} = 9.71$, $p < 0.01$, $n = 51$), but not juveniles ($F_{1,34} = 0.54$, $p = 0.468$, $n = 62$). There was no relationship between population growth and the intensity of *Isospora* infection ($F_{1,14} = 0.089$, $p = 0.769$, $n = 76$), but population growth was negatively related to *Isospora* prevalence among juveniles ($F_{1,34} = 4.74$, $p = 0.036$, $n = 48$), but not adults ($F_{1,42} = 0.006$, $p = 0.938$, $n = 28$; table 3).

## 3.4. Effect of parasite infection intensity on individual host survival

The intensity of *Plasmodium* infection was included in six of the seven most parsimonious survival models (which together accounted for 79% of total AICc weights), while age was included in five of the seven models (electronic supplementary material, table S4). In all six models, sparrow survival was negatively related to the intensity of *Plasmodium* infection. This negative relationship was statistically significant for both adult and juvenile sparrows, although intensity of infection in adults was lower than in juveniles (figure 3 and table 4).

Two of the six best models included *Atoxoplasma* infection intensity as a survival covariate but these received relatively low support (18% of total AICc weights, electronic supplementary material, table S5); the relationship between intensity of infection and survival was not significant in either model ($\beta = 0.034 \pm 0.115$ s.e. and $\beta = 0.035 \pm 0.114$ s.e.).

## 3.5. Impact of NO$_2$ pollution on house sparrow population growth, parasite prevalence and infection intensity

No significant relationship was determined between sparrow population growth rate and the environmental concentration of NO$_2$ (electronic supplementary material, table S6). Also, none of the relationships between environmental NO$_2$ concentration and parasite prevalence or infection intensity were significant, with the exception of a negative relationship with *Isospora* prevalence (table 5).

## 4. Discussion

Knowledge of parasite prevalence and intensity can provide important insights into the dynamics of host–parasite interactions [45]. We examined relationships between a variety of microparasite infections and the demography of suburban sparrows. We found low prevalences of gastro-enteric

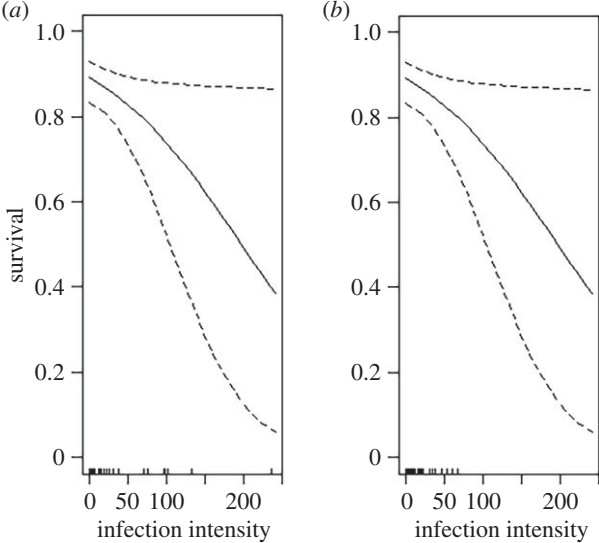

**Figure 3.** Relationship between intensity of *Plasmodium relictum* infection and monthly survival of (*a*) juvenile and (*b*) adult house sparrows (*Passer domesticus*), illustrating the age × plasmodium survival model with independent slopes (Model 2 electronic supplementary material, table S4). Ticks on the *x*-axis indicate infection intensity per individual sparrow.

**Table 5.** Summary of the results on the association between environmental concentrations of $NO_2$ and prevalence and infection intensity of *Plasmodium*, *Atoxoplasma* and *Isospora* at the 11 study sites. Parasite infection prevalence was calculated as the number infected/number sampled house sparrows, while the intensity of infection was calculated as the number of parasites detected per individual host. Significant results at 95% level in italics.

| parasite | measure | age | estimate | *t* | *p*-value |
|---|---|---|---|---|---|
| plasmodium | prevalence | both | 0.035 ± 0.041 | 0.881 | 0.378 |
| plasmodium | prevalence | adults | 0.067 ± 0.05 | 1.339 | 0.181 |
| plasmodium | prevalence | juveniles | −0.024 ± 0.067 | −0.362 | 0.718 |
| plasmodium | intensity | both | 0.002 ± 0.005 | 0.391 | 0.705 |
| plasmodium | intensity | adults | 0.002 ± 0.014 | 0.118 | 0.909 |
| plasmodium | intensity | juveniles | 0.005 ± 0.011 | 0.423 | 0.685 |
| atoxoplasma | prevalence | both | 0.041 ± 0.037 | 1.113 | 0.266 |
| atoxoplasma | prevalence | adults | 0.093 ± 0.048 | 1.923 | 0.055 |
| atoxoplasma | prevalence | juveniles | −0.032 ± 0.059 | −0.542 | 0.588 |
| atoxoplasma | intensity | both | 0.002 ± 0.014 | 0.118 | 0.909 |
| atoxoplasma | intensity | adults | 0.007 ± 0.013 | 0.591 | 0.571 |
| atoxoplasma | intensity | juveniles | −0.072 ± 0.043 | −1.676 | 0.138 |
| isospora | prevalence | both | −0.215 ± 0.102 | −2.109 | *0.035* |
| isospora | prevalence | adults | −0.334 ± 0.223 | −1.497 | 0.134 |
| isospora | prevalence | juveniles | −0.203 ± 0.127 | −1.599 | 0.110 |
| isospora | intensity | both | −0.116 ± 0.144 | −0.802 | 0.507 |
| isospora | intensity | adults | n.a. | n.a. | n.a. |
| isospora | intensity | juveniles | 0.044 ± 0.117 | 0.375 | 0.772 |

helminths and bacterial pathogens but moderate to high prevalences of three protozoan parasites (*Isospora*, *Atoxoplasma* and *Plasmodium*). The prevalence of *Plasmodium* infection (50–100%), detected using light microscopy, was higher than has previously been reported for any species of free-living wild bird in Northern Europe [46–49].

We did not find a consistent or strong relationship between prevalence or intensity of infection with *Isospora* and population growth rate; sparrows with *Isospora* infection were detected only at four sites. Only one individual was positive for both *Isospora* and *Atoxoplasma*, which suggests that, in our study populations, the two infections were distinct.

Although we found high prevalences of *Plasmodium* in our study populations, there was no association between prevalence and local sparrow population growth. Infection intensity, however, was significantly higher in juveniles (although not in adults) in declining populations. Higher levels of infection intensity among juvenile sparrows has been documented before in France (e.g. [50]) but this is the first time that higher infection intensity has been reported to be related to lower population growth. Furthermore, *Plasmodium* infection intensity was a negative predictor of individual overwinter survival of both adult and juvenile sparrows. This association is likely to be conservative as some sparrows with intense infections may have died before we were able to capture them. Demographic studies in London indicate that sparrow population declines are associated with reduced recruitment of juveniles, probably linked to lower overwinter survival [19]. At the UK scale, demographic data also highlight low juvenile survival as a candidate demographic driver of the wider population decline [20]. Our findings are consistent with high intensities of *Plasmodium* infection reducing overwinter survival (of adults and juveniles) and the recruitment of first-time breeders, leading to colony declines. Because sparrows have low natal and breeding dispersal [25,27], reduced overwinter survival is unlikely to be buffered by immigration. Our findings, therefore, implicate *Plasmodium* infection as a factor linked to low survival and population declines among suburban sparrows in London.

Sequenced amplicons from randomly chosen samples of blood from our study were all identified as being from *P. relictum* SGS1 or *P. relictum* GRW11. These are globally widespread, generalist lineages and are likely to have been native to the UK (and to the sparrow) long before the onset of population declines [51], so why might this parasite now be affecting the demography of urban sparrows in the UK? Since we show that intensity of infection, rather than simply infection *per se*, is associated with reduced survival and population decline, if *Plasmodium* has caused the recent population decline, we would expect that the average intensity of infection has risen. Such an increase in intensity of infection could potentially be caused by (1) a change in the host's ability to control infection, as might occur if sparrows were immunosuppressed, (2) a change in the virulence of the parasite or (3) a change in the infection dynamics of the host–parasite system.

Host immunosuppression can be caused by viral or other infections, toxins, malnutrition or stress [52,53]. Bichet *et al*. [22] found an association between trace metals and the prevalence of *Plasmodium* in house sparrows and suggested that pollution might compromise the host immune system. In the current study, however, only infection intensity of *Plasmodium* (and not *Atoxoplasma*) was associated with reduced survival and, while other parasites were detected, there was no indication that their prevalence or intensity was related to host survival or local population trend (with the possible exception of *Isospora*, but see discussion above). A small body of evidence has recently shown that co-infection with parasites does not always have negative additive effects. Despite some evidence that co-infection can reduce the intensity of *Plasmodium* infections in primates [54,55], the absence of other parasites impacting sparrow demographics suggests immunosuppression was not a factor in the sparrows we studied.

If a change in parasite virulence, perhaps through genetic mutation or introduction of a hypervirulent lineage, had occurred, we would have expected to find a single strain of *Plasmodium* at the declining sites and this strain to be absent from stable or increasing sparrow populations. Instead, we found two separate *Plasmodium* lineages infecting sparrows at both declining and non-declining sites, although not enough samples were sequenced to test for an association between any particular lineage and individual host survival or demographic trend.

Increased infection intensity could also arise from a change in infection dynamics, for example, through increased exposure of the host to the parasite via a higher abundance of, or a higher efficiency of parasite inoculation by, infected mosquitoes. The former is likely to be driven by local environmental conditions (e.g. warm and wet conditions favouring mosquito reproduction and development) but we have no data on vector abundance at our study sites. It has been hypothesized that *Plasmodium* prevalence will increase across Northern Europe due to climate warming [46], and that climate change will influence avian malaria infection rates through increased parasite and vector abundance and altered mosquito distributions [56]. The role of the initial infection dose in the subsequent intensity of infection, however, is unclear. Some authors have reported that *Plasmodium* infection intensity is unrelated to the inoculation dose in sparrows, European starlings (*Sturnus vulgaris*) [57] or domestic chickens (*Gallus gallus domesticus*) [58]. However, using infected mosquitoes

rather than direct inoculation as the vector of the malarian parasite in their experiment, Atkinson *et al.* [6] found an association between the intensity of infection between day 8 and day 20 post infection with the infection dose in i'iwi (*Vestiaria coccinea*), especially if bitten by multiple infected mosquitoes. As resistance to *Plasmodium* has been found to have a genetic basis in sparrows [47], increased intensities of infection might have a greater fitness impact on populations where exposure rates were historically lower.

As air pollution has been suggested as a possible factor in the decline of the London sparrow population [15,17], we tested for associations between local $NO_2$ concentrations and our measures of parasite infection. We found no significant relationship between the level of local $NO_2$ air pollution and population growth rate (electronic supplementary material, table S6), or with prevalence or intensity of *Plasmodium* or *Atoxoplasma* infection (table 5). There was a negative relationship with the prevalence of *Isospora*, which was lower in more polluted areas (table 5), a result which is difficult to interpret ecologically.

Sparrow populations are highly spatially structured with limited dispersal between sub-populations, especially after the initial post-fledging period [25,59], and urban/suburban populations have been shown to be less genetically diverse than rural ones [60]. Thus, local population extinctions might not be balanced by recolonization events, especially when neighbouring populations are declining. Although we were unable to test whether dispersal rates in our populations had changed, the mean distance from ringing location of house sparrows ringed in the UK and subsequently found dead elsewhere, a measure of dispersal distance, decreased between 1970 (15.0 km $\pm$ 1.4, mean $\pm$ 1 s.e.) and 2010 (2.5 km $\pm$ 0.6, $t_1 = -5.83$, $p < 0.001$; British Trust for Ornithology, 2018, unpublished data). Therefore, even mainland populations of widespread sedentary species may become isolated and suffer from an island effect. Low dispersal could help facilitate a parasite-driven decline through a combination of low recruitment and genetic bottlenecks reducing resistance to disease (such as avian malaria). It has been shown in the wild that inbred individuals are more susceptible to parasites [61], and that in house sparrows there can be population-specific variations in MHC alleles associated with the risk of infection with *P. relictum* [62]. Inbreeding coefficient has been associated with a high level of homozygosity in sparrows [63], therefore individuals with lower heterozygosity may be less able to mount an effective immune response [64].

## 5. Conclusion

The underlying causes of house sparrow population declines remain unknown. Here, we show that in declining sparrow populations across suburban London, sparrows have a higher intensity of *Plasmodium* infection than those in non-declining populations, and that the intensity of that infection is associated with reduced overwinter survival of both juveniles and adults. Why this is occurring is unclear, but changes in host–parasite infection dynamics, possibly resulting from climatic warming and/or changes in vector distributions, could underlie this. Post-juvenile dispersal distance may have decreased over recent years reducing the ability for recruitment losses to be recouped through immigration and this could be exacerbating the population impacts of the *Plasmodium*-mediated mortality we observed. The extent of the impact of infectious disease on biodiversity has only recently been appreciated [2,65], but is hampered by a lack of baseline data. Improved surveillance, knowledge and understanding of perturbations to host–parasite systems is critical for effective conservation management [66]. We suggest that infectious diseases should be considered when evaluating potential causes of population decline, including of widespread and common species.

Ethics. This research project was approved by the Zoological Society of London's Ethics Committee (WLE447). Blood sampling was conducted under a Home Office license.

Data accessibility. The datasets and codes supporting this article have been uploaded as part of the electronic supplementary material.

Authors' contributions. A.A.C. conceived the study. A.A.C., M.B., D.D. and W.P. formulated the ideas. D.D. performed fieldwork. D.D. and A.C. performed laboratory work. D.D., R.A.R. and J.M.R. analysed and modelled the data. D.D. wrote the first draft of the manuscript, and all authors contributed to revisions.

Competing interests. We declare we have no competing interests.

Funding. This study was funded by NERC and the Royal Society for the Protection of Birds. A.A.C. was supported by a Royal Society Wolfson research merit award.

Acknowledgements. We thank Shaheed MacGregor, Shinto John, Sreejith Radhakrishnan, Chris Orsman, William Haines, Nancy Ockendon, John Mallord, Michael Clarkson, Alison Johnston and Mike Hart for technical assistance, Loris Dadam for comments on an earlier version of the manuscript, the Laboratory of Enteric Pathogens, Public Health

England for confirmatory tests on suspected *Salmonella* sp. isolates, and all the householders who permitted access to their gardens to count and sample sparrows. We thank also the three anonymous referees for their comments on the manuscript.

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
