## [Reviewer comments · Royal Society Open Science]

Review History

RSOS-182197.R0 (Original submission)

Review form: Reviewer 1

Is the manuscript scientifically sound in its present form?

Yes

Are the interpretations and conclusions justified by the results?

Yes

Is the language acceptable?

Yes

Is it clear how to access all supporting data?

Yes

Do you have any ethical concerns with this paper?

No

Have you any concerns about statistical analyses in this paper?

I do not feel qualified to assess the statistics

Recommendation?

Accept with minor revision (please list in comments)

Comments to the Author(s)

The reviewed manuscript presents interesting and novel findings on the influence of pathogens such as protozoan parasites (Haemosporida and Coccidia, Eucoccidiorida), bacteria, and helminths on populations declines of house sparrows (*Passer domesticus*) in Lindon, UK. The paper is well written and well structured. However, I think that section "1. Introduction" have to be expanded and more informative. Perhaps the following studies worth citing and discussion: Loiseau, C., Zoorob, R., Robert, A., Chastel, O., Julliard, R. & Sorci, G. (2011) *Plasmodium relictum* infection and MHC diversity in the house sparrow (*Passer domesticus*). *Proceedings. Biological sciences / The Royal Society* 278, 1264-1272.

Bichet, C., Sorci, G., Robert, A., Julliard, R., Lendvai, Á.Z., Chastel, O., Garnier, S. & Loiseau, C. (2014) Epidemiology of *Plasmodium relictum* infection in the house sparrow. *Journal of Parasitology* 100, 59-65.

Marzal, A., Ricklefs, R.E., Valkiūnas, G., Albayrak, T., Arriero, E., Bonneaud, C., Czirják, G. a, Ewen, J., Hellgren, O., Hořáková, D., Iezhova, T.A., Jensen, H., Križanauskienė, A., Lima, M.R., de Lope, F., Magnussen, E., Martin, L.B., Møller, A.P., Palinauskas, V., Pap, P.L., Pérez-Tris, J., Sehgal, R.N.M., Soler, M., Szölloši, E., Westerdahl, H., Zetindjiev, P. & Bensch, S. (2011) Diversity, loss, and gain of malaria parasites in a globally invasive bird. *PloS one* 6, e21905.

For me was also quite difficult to move between methodology in the section "2. Methods" and supplementary materials S1-S4 in order to get familiar with it. I would recommend if possible gathering them together.

Another questionable part of the methodology is how *Atoxoplasma* and *Isospora* species was differentiated despite the discussion on L210-216. Since *Atoxoplasma* was suggested as junior synonym of *Isospora*, how they were separated based on the blood smears or fecal samples? The scientific terminology was not always appropriately used. For example L 161 "intensity of parasitism" Perhaps you mean intensity of infection or parasitaemia? For blood parasites "parasitaemia" is more commonly used. I would recommend the following paper to solve similar issues:

Bush, A.O., Lafferty, K.D., Lotz, J.M. & Shostak, A.W. (1997) Parasitology meets ecology on its own terms: *Macrogolis* et al. Revisited. *Journal of Parasitology* 83, 575-583.

Line 268 "inoculation dose", please note that infection initiated via artificial inoculation with infected blood and via vector bite are different techniques and it is not correct to compare them equally! On L 272 you have multiple inoculations with sporozoites of *P. relictum*, which is not the same as studies 45 and 46. In my opinion it worth discussing the abundance of vectors of *P. relictum*, *Culex pipiens* if you have such data for the different sparrow populations in your study rather than compare with experimental infections initiated with inoculation of infected blood.

Review form: Reviewer 2**Is the manuscript scientifically sound in its present form?**

Yes

Are the interpretations and conclusions justified by the results?

Yes

Is the language acceptable?

Yes

Is it clear how to access all supporting data?

Yes

Do you have any ethical concerns with this paper?

No

Have you any concerns about statistical analyses in this paper?

I do not feel qualified to assess the statistics

Recommendation?

Accept with minor revision (please list in comments)

Comments to the Author(s)

Abstract:

A couple wording suggestions:

Line 22: "Populations of house sparrow (*Passer domesticus*) have declined in many European towns and cities, but the causes of these declines remain unclear."

Line 26: "*Plasmodium relictum* infection was found at higher prevalences (averaging 74%) in suburban London house sparrows than previously recorded in any wild bird population in Northern Europe."

Line 31: "Our findings indicate that *P. relictum* infection may be influencing house sparrow population dynamics in suburban areas."

Introduction:

Starts well with broad themes, but the focus needs to narrow down to the present study. The third sentence of the first paragraph (Line 42) is vague, what are the potential demographic impacts of parasites on birds?

Also, avian malaria is in the paper's title, but there is no mention of it in the Introduction.

Introduce *Plasmodium* perhaps around Line 52 when mentioning parasites are generally understudied as a potential cause of avian population declines.

Wording:

Line 45: "The house sparrow (*Passer domesticus*, hereafter sparrow) used to be one of the most common species of bird in the United Kingdom (UK) ..."

Methods

Line 64: States the study was conducted between November 2006 and September 2009 but section (b) of Methods says surveys of male sparrows were completed between 2005 and 2009. Does it matter for this study that annual surveys of male sparrows were conducted a year before this study began?

Line 67: Is 4 km a significant distance for house sparrow populations or is this just a minimum distance between study populations?

Line 87: Was data from the "pilot study" used in the analysis?

Line 90: Write out full years: 2007/2008 and 2008/2009

Lines 96-97: You state you identified the relative abundance of parasites in each faecal sample (but not swab). What are the swabs you are referring to here?

Line 125: Wording - "Separate analyses were conducted first based on infection rates of all birds, and then of adults and juveniles separately. Too few individuals were sexed to allow sex-specific

analyses.

Line 137: Add “n =” in front of 117 when reporting sample size

Lines 166-169: Run-on sentence. Each statement here should be in a single sentence.

Section (c): Report sample size with all statistics

Results:

Line 176: You report there was no association between prevalence of Plasmodium and local sparrow population growth in the Discussion but this should also be reported here in the Results.

Line 228: You switch from “juveniles and adults” to “adults and first years”. Stay consistent.

Line 235: Plasmodium lineages cannot be “globally widespread” and “endemic to the UK”.

Perhaps use “native” instead of “endemic” here?

Discussion:

In the Introduction you mentioned the multiple hypotheses put forth to explain house sparrow decline. It might be informative to discuss those studies in the Discussion, especially if some of those factors may be contributing to the overall decline along with avian malaria.

Line 245: Bichet et al.?

Line 269: Are you referring to house sparrows or more broadly to sparrows as a family? Either way, specify here.

Line 270: Full species name is European starling.

Line 271: Atkinson et al.?

Line 277: This paragraph does not fit well here without prior introduction. This information either needs to be presented in the Methods and Results sections, or not at all. It might be more appropriate as a stand-alone publication such as a “Short Communication”.

Line 296: Wording – “Inbreeding coefficient has been associated with a high level of homozygosity in sparrows [50], therefore individuals with lower heterozygosity may be less able to mount an immune response [51].

Tables and Figures:

Figure 1: Figures should be able to stand-alone, title needs more description – “Map of house sparrow (*Passer domesticus*) study sites in London, indicating whether or not counts of territorial male house sparrows showed a declining trend over the period 2005-2009”.

Figure 2: Figure should stand-alone. Report that parasite prevalence and intensity was measured in house sparrows (*Passer domesticus*). Report what the line represents in figure a) in the description. Also report that each symbol represents a distinct house sparrow population. This might read better as two separate figures.

Figure 3: “Relationship between intensity of *Plasmodium relictum* infection and monthly survival of (a) juvenile and (b) adult house sparrows (*Passer domesticus*), illustrating the age*plasmodium survival model with independent slopes (Model 2 Table S4). Ticks on the x-axis indicate infection intensity per individual sparrow.”

Table 1: Use the percent sign when reporting each percent. You’ve placed the percent sign in the column titles but I think it would be clearer to write it out next to each percentage.

Review form: Reviewer 3

Is the manuscript scientifically sound in its present form?

Yes

Are the interpretations and conclusions justified by the results?

Yes

Is the language acceptable?

Yes

Is it clear how to access all supporting data?

Yes

Do you have any ethical concerns with this paper?

No

Have you any concerns about statistical analyses in this paper?

Yes

Recommendation?

Major revision is needed (please make suggestions in comments)

Comments to the Author(s)

Major comments:

L. 28-30 – That other parasites were less prevalent and no relationship with house sparrow survival, etc. assumes that co-infection is always ‘negative’ where some (limited) literature suggests that the negative effects of co-infection do not worsen additively with each new parasite infection. Consider addressing this briefly in the Discussion paragraph starting at L. 244.

Analyses – I appreciate the care that was taken in investigating the effects of parasitism on house sparrows. However, several issues emerged that should be considered to make the analytical approach and inferences derived from it stronger. These include addressing the following:

A) L.122-123 – How were the analyses (models) weighted by prevalence or intensity? This was confusing to me.

B) L. 130 – What was the CJS ‘extension’ that was used? (May be simply “CJS model”)

C) CJS modeling approach – If I understood correctly, the mark-recapture analyses had two phases. One was to ask about the ecological factors associated with apparent survival and detection. Why take this stepped approach? Consider providing additional detail and reasoning for this approach directly in the MS. Related, as I understand it, the infection intensity values were those determined using the blood sample taken when the birds were first caught and banded. Couldn’t infection intensity change over the course of the study? It IS interesting that survival varies with infection intensity and age but are infection intensity and age correlated? If so, the inferences about age and intensity get complicated. Also, what about the simple relationship between infection status at first capture and apparent survival? This seems like an important question that is not addressed but could be easily.

D) Mixing paradigms – Avoid mixing the information-theoretic (IT) approaches used to report the capture-recapture model selection results with null hypothesis testing that is used for the population growth analyses. Consider reworking the population growth analyses in an IT framework; this allows discussion of the weight of evidence for competing models and lets the factors of interest ‘duke it out’ with respect to which one or ones best explain the variation in the data. This approach also allows a deeper analysis of the combinations of factors – including infection intensity - that might be associated with changes in population size (is it size or is it growth like lambda?) A minor point – significance wording is not needed when reporting effect sizes from the ‘best model’ under an IT framework – that veers into mixing paradigms within the same analysis instead of just within the same paper. The NO2 analyses could also be done in an IT framework. Consider consulting Anderson 2007 - Model Based Inference in the Life Sciences: A Primer on Evidence and associated material for more on analyses in an IT framework.

Minor edits to consider:

L. 30 - Add "was shown" after "abundance"

L. 50 - Add "negative" before "impacts of pollutants"

L.57 - 73% here is different than the value cited in the Abstract (72%)

L. 68 - consider replacing "travel" with "disperse"; when I first read it I thought to myself "They travel a lot through flight...maybe kilometers in a day"

L.189 - "AIC support" would be better stated as "Akaike weight" or "AICc weight" (the AICc is the small sample size corrected form of AIC and is often used almost by default because AIC = AICc when sample sizes are large)

L. 250 - correct spelling to "exception"

L. 307 - consider replacing "reduced" with "decreased"

Decision letter (RSOS-182197.R0)

27-Feb-2019

Dear Dr Dadam,

The editors assigned to your paper ("Avian malaria-mediated population decline of a widespread iconic bird species") have now received comments from reviewers. We would like you to revise your paper in accordance with the referee and Associate Editor suggestions which can be found below (not including confidential reports to the Editor). Please note this decision does not guarantee eventual acceptance.

Please submit a copy of your revised paper before 22-Mar-2019. Please note that the revision deadline will expire at 00.00am on this date. If we do not hear from you within this time then it will be assumed that the paper has been withdrawn. In exceptional circumstances, extensions may be possible if agreed with the Editorial Office in advance. We do not allow multiple rounds of revision so we urge you to make every effort to fully address all of the comments at this stage. If deemed necessary by the Editors, your manuscript will be sent back to one or more of the original reviewers for assessment. If the original reviewers are not available, we may invite new reviewers.

- Data accessibility

<http://datadryad.org/submit?journalID=RSOS&manu=RSOS-182197>

- Competing interests

- Authors' contributions

- Acknowledgements

- Funding statement

on behalf of Dr Cynthia Downs (Associate Editor) and Professor Kevin Padian (Subject Editor)
openscience@royalsociety.org

Associate Editor's comments (Dr Cynthia Downs):

The manuscript "Avian malaria-mediated population decline of a widespread iconic bird species" was reviewed by 3 expert reviewers and myself. The study presented in the manuscript uses traditional population ecology techniques and demographic modeling to assess the influence of parasites on the decline of house sparrow populations in Lindon, UK. The finding that infection intensity of *Plasmodium relictum* was negatively related to survival rates of juvenile and adult sparrows and population growth rate is exciting. The manuscript was generally well written and enjoyable to read.

I suggest extending the introduction to provide more context for how the current study fits into the literature as suggested by two of the reviewers. I also recommend addressing the comments made by reviewer 3 about the statistical modeling approach. Particularly with regards to clarifying the statistical methods used and reviewer 3's suggestion to apply an information-theoretic approach to analyse population growth. In addition, the models used to develop the model-averaged best model for survival included the model set within 2 delta AICs of the top model. It is more common to use the model set within 4 delta AICs of the top model. Please explore this more conservative approach to ensure that it does not change the interpretation of the data. Also, please address the suggestions from the reviewers about clarifying methods that are important for interpreting the scientific soundness of the study presented.

This manuscript has the potential inform our understanding of whether avian malaria has contributed to the decline of house sparrows in the United Kingdom.

Comments to Author:

Reviewers' Comments to Author:

Reviewer: 1

Comments to the Author(s)

The reviewed manuscript presents interesting and novel findings on the influence of pathogens such as protozoan parasites (Haemosporida and Coccidia, Eucoccidiorida), bacteria, and helminths on populations declines of house sparrows (*Passer domesticus*) in Lindon, UK. The paper is well written and well structured. However, I think that section "1. Introduction" have to be expanded and more informative. Perhaps the following studies worth citing and discussion: Loiseau, C., Zoorob, R., Robert, A., Chastel, O., Julliard, R. & Sorci, G. (2011) *Plasmodium relictum* infection and MHC diversity in the house sparrow (*Passer domesticus*). *Proceedings. Biological sciences / The Royal Society* 278, 1264–1272.

Bichet, C., Sorci, G., Robert, A., Julliard, R., Lendvai, Á.Z., Chastel, O., Garnier, S. & Loiseau, C. (2014) Epidemiology of *Plasmodium relictum* infection in the house sparrow. *Journal of Parasitology* 100, 59–65.

Marzal, A., Ricklefs, R.E., Valkiūnas, G., Albayrak, T., Arriero, E., Bonneaud, C., Czirják, G. a, Ewen, J., Hellgren, O., Hořáková, D., Iezhova, T.A., Jensen, H., Križanauskienė, A., Lima, M.R., de Lope, F., Magnussen, E., Martin, L.B., Møller, A.P., Palinauskas, V., Pap, P.L., Pérez-Tris, J., Sehgal, R.N.M., Soler, M., Szölloši, E., Westerdahl, H., Zetindjiev, P. & Bensch, S. (2011) Diversity, loss, and gain of malaria parasites in a globally invasive bird. *PloS one* 6, e21905.

For me was also quite difficult to move between methodology in the section "2. Methods" and supplementary materials S1-S4 in order to get familiar with it. I would recommend if possible gathering them together.

Another questionable part of the methodology is how *Atoxoplasma* and *Isospora* species was differentiated despite the discussion on L210-216. Since *Atoxoplasma* was suggested as junior synonym of *Isospora*, how they were separated based on the blood smears or fecal samples?

The scientific terminology was not always appropriately used. For example L 161 "intensity of parasitism" Perhaps you mean intensity of infection or parasitaemia? For blood parasites "parasitaemia" is more commonly used. I would recommend the following paper to solve similar issues:

Bush, A.O., Lafferty, K.D., Lotz, J.M. & Shostak, A.W. (1997) Parasitology meets ecology on its own terms: Macrogolis et al. Revisited. *Journal of Parasitology* 83, 575–583.

Line 268 "inoculation dose", please note that infection initiated via artificial inoculation with infected blood and via vector bite are different techniques and it is not correct to compare them equally! On L 272 you have multiple inoculations with sporozoites of *P. relictum*, which is not the same as studies 45 and 46. In my opinion it worth discussing the abundance of vectors of *P. relictum*, *Culex pipiens* if you have such data for the different sparrow populations in your study rather than compare with experimental infections initiated with inoculation of infected blood.

Reviewer: 2

Comments to the Author(s)

Abstract:

A couple wording suggestions:

Line 22: "Populations of house sparrow (*Passer domesticus*) have declined in many European towns and cities, but the causes of these declines remain unclear."

Line 26: "*Plasmodium relictum* infection was found at higher prevalences (averaging 74%) in suburban London house sparrows than previously recorded in any wild bird population in Northern Europe."

Line 31: "Our findings indicate that *P. relictum* infection may be influencing house sparrow population dynamics in suburban areas."

Introduction:

Starts well with broad themes, but the focus needs to narrow down to the present study. The third sentence of the first paragraph (Line 42) is vague, what are the potential demographic impacts of parasites on birds?

Also, avian malaria is in the paper's title, but there is no mention of it in the Introduction. Introduce *Plasmodium* perhaps around Line 52 when mentioning parasites are generally understudied as a potential cause of avian population declines.

Wording:

Line 45: "The house sparrow (*Passer domesticus*, hereafter sparrow) used to be one of the most common species of bird in the United Kingdom (UK) ..."

Methods

Line 64: States the study was conducted between November 2006 and September 2009 but section (b) of Methods says surveys of male sparrows were completed between 2005 and 2009. Does it matter for this study that annual surveys of male sparrows were conducted a year before this study began?

Line 67: Is 4 km a significant distance for house sparrow populations or is this just a minimum distance between study populations?

Line 87: Was data from the “pilot study” used in the analysis?

Line 90: Write out full years: 2007/2008 and 2008/2009

Lines 96-97: You state you identified the relative abundance of parasites in each faecal sample (but not swab). What are the swabs you are referring to here?

Line 125: Wording – “Separate analyses were conducted first based on infection rates of all birds, and then of adults and juveniles separately. Too few individuals were sexed to allow sex-specific analyses.

Line 137: Add “n =” in front of 117 when reporting sample size

Lines 166-169: Run-on sentence. Each statement here should be in a single sentence.

Section (c): Report sample size with all statistics

Results:

Line 176: You report there was no association between prevalence of Plasmodium and local sparrow population growth in the Discussion but this should also be reported here in the Results.

Line 228: You switch from “juveniles and adults” to “adults and first years”. Stay consistent.

Line 235: Plasmodium lineages cannot be “globally widespread” and “endemic to the UK”. Perhaps use “native” instead of “endemic” here?

Discussion:

In the Introduction you mentioned the multiple hypotheses put forth to explain house sparrow decline. It might be informative to discuss those studies in the Discussion, especially if some of those factors may be contributing to the overall decline along with avian malaria.

Line 245: Bichet et al.?

Line 269: Are you referring to house sparrows or more broadly to sparrows as a family? Either way, specify here.

Line 270: Full species name is European starling.

Line 271: Atkinson et al.?

Line 277: This paragraph does not fit well here without prior introduction. This information either needs to be presented in the Methods and Results sections, or not at all. It might be more appropriate as a stand-alone publication such as a “Short Communication”.

Line 296: Wording – “Inbreeding coefficient has been associated with a high level of homozygosity in sparrows [50], therefore individuals with lower heterozygosity may be less able to mount an immune response [51].

Tables and Figures:

Figure 1: Figures should be able to stand-alone, title needs more description – “Map of house sparrow (*Passer domesticus*) study sites in London, indicating whether or not counts of territorial male house sparrows showed a declining trend over the period 2005-2009”.

Figure 2: Figure should stand-alone. Report that parasite prevalence and intensity was measured in house sparrows (*Passer domesticus*). Report what the line represents in figure a) in the description. Also report that each symbol represents a distinct house sparrow population. This might read better as two separate figures.

Figure 3: “Relationship between intensity of *Plasmodium relictum* infection and monthly survival of (a) juvenile and (b) adult house sparrows (*Passer domesticus*), illustrating the age*plasmodium survival model with independent slopes (Model 2 Table S4). Ticks on the x-axis indicate infection intensity per individual sparrow.”

Table 1: Use the percent sign when reporting each percent. You've placed the percent sign in the column titles but I think it would be clearer to write it out next to each percentage.

Reviewer: 3

Comments to the Author(s)

Major comments:

L. 28-30 - That other parasites were less prevalent and no relationship with house sparrow survival, etc. assumes that co-infection is always 'negative' where some (limited) literature suggests that the negative effects of co-infection do not worsen additively with each new parasite infection. Consider addressing this briefly in the Discussion paragraph starting at L. 244.

Analyses - I appreciate the care that was taken in investigating the effects of parasitism on house sparrows. However, several issues emerged that should be considered to make the analytical approach and inferences derived from it stronger. These include addressing the following:

A) L.122-123 - How were the analyses (models) weighted by prevalence or intensity? This was confusing to me.

B) L. 130 - What was the CJS 'extension' that was used? (May be simply "CJS model")

C) CJS modeling approach - If I understood correctly, the mark-recapture analyses had two phases. One was to ask about the ecological factors associated with apparent survival and detection. Why take this stepped approach? Consider providing additional detail and reasoning for this approach directly in the MS. Related, as I understand it, the infection intensity values were those determined using the blood sample taken when the birds were first caught and banded. Couldn't infection intensity change over the course of the study? It IS interesting that survival varies with infection intensity and age but are infection intensity and age correlated? If so, the inferences about age and intensity get complicated. Also, what about the simple relationship between infection status at first capture and apparent survival? This seems like an important question that is not addressed but could be easily.

D) Mixing paradigms - Avoid mixing the information-theoretic (IT) approaches used to report the capture-recapture model selection results with null hypothesis testing that is used for the population growth analyses. Consider reworking the population growth analyses in an IT framework; this allows discussion of the weight of evidence for competing models and lets the factors of interest 'duke it out' with respect to which one or ones best explain the variation in the data. This approach also allows a deeper analysis of the combinations of factors - including infection intensity - that might be associated with changes in population size (is it size or is it growth like lambda?) A minor point - significance wording is not needed when reporting effect sizes from the 'best model' under an IT framework - that veers into mixing paradigms within the same analysis instead of just within the same paper. The NO2 analyses could also be done in an IT framework. Consider consulting Anderson 2007 - Model Based Inference in the Life Sciences: A Primer on Evidence and associated material for more on analyses in an IT framework.

Minor edits to consider:

L. 30 - Add "was shown" after "abundance"

L. 50 - Add "negative" before "impacts of pollutants"

L.57 - 73% here is different than the value cited in the Abstract (72%)

L. 68 - consider replacing "travel" with "disperse"; when I first read it I thought to myself "They travel a lot through flight...maybe kilometers in a day"

L.189 - "AIC support" would be better stated as "Akaike weight" or "AICc weight" (the AICc is the small sample size corrected form of AIC and is often used almost by default because AIC = AICc when sample sizes are large)

L. 250 - correct spelling to "exception"

L. 307 - consider replacing "reduced" with "decreased"

Author's Response to Decision Letter for (RSOS-182197.R0)

See Appendix A.

RSOS-182197.R1 (Revision)

Review form: Reviewer 1

Is the manuscript scientifically sound in its present form?

Yes

Are the interpretations and conclusions justified by the results?

Yes

Is the language acceptable?

Yes

Is it clear how to access all supporting data?

Yes

Do you have any ethical concerns with this paper?

No

Have you any concerns about statistical analyses in this paper?

I do not feel qualified to assess the statistics

Recommendation?

Accept with minor revision (please list in comments)

Comments to the Author(s)

The manuscript was significantly improved from its previous version. However, I think the Introduction is still too short and lacking of information. In my opinion, some parts from the large Discussion might be modified and moved to the Introduction, as for example the last paragraph of the Discussion (L408-432).

The terminology use is still not appropriate on some places (L58, L62). Please be aware that "parasitaemia " and "infection intensity" have the same meaning in parasitology. I would recommend to stay consistent and use one of the terms. For blood parasites "parasitaemia" is more commonly used.

Why I could not find supplementary Table 2? Is it wrongly entitled as TableS 3?

Review form: Reviewer 2

Is the manuscript scientifically sound in its present form?

Yes

Are the interpretations and conclusions justified by the results?

Yes

Is the language acceptable?

Yes

Is it clear how to access all supporting data?

Yes

Do you have any ethical concerns with this paper?

No

Have you any concerns about statistical analyses in this paper?

I do not feel qualified to assess the statistics

Recommendation?

Accept as is

Comments to the Author(s)

Thank you for taking the time to thoughtfully address and respond to reviewer comments and suggestions. Looking forward to seeing this manuscript published.

Decision letter (RSOS-182197.R1)

29-Apr-2019

Dear Dr Dadam:

Manuscript ID RSOS-182197.R1 entitled "Avian malaria-mediated population decline of a widespread iconic bird species" which you submitted to Royal Society Open Science, has been reviewed. The comments of the reviewer(s) are included at the bottom of this letter.

Please submit a copy of your revised paper before 22-May-2019. Please note that the revision deadline will expire at 00.00am on this date. If we do not hear from you within this time then it will be assumed that the paper has been withdrawn. In exceptional circumstances, extensions may be possible if agreed with the Editorial Office in advance. We do not allow multiple rounds of revision so we urge you to make every effort to fully address all of the comments at this stage. If deemed necessary by the Editors, your manuscript will be sent back to one or more of the original reviewers for assessment. If the original reviewers are not available we may invite new reviewers.

When submitting your revised manuscript, you must respond to the comments made by the referees and upload a file "Response to Referees" in "Section 6 - File Upload". Please use this to

document how you have responded to the comments, and the adjustments you have made. In order to expedite the processing of the revised manuscript, please be as specific as possible in your response.

- Ethics statement

- Data accessibility

- Competing interests

- Authors' contributions

- Acknowledgements

- Funding statement

Please do be aware that, ordinarily, multiple rounds of major revision are not considered by the

journal - thus, it is a recognition of the strong efforts you have made that the Editors are allowing this second round of revision. You should note that further revisions will not be possible.

on behalf of Dr Cynthia Downs (Associate Editor) and Kevin Padian (Subject Editor)
openscience@royalsociety.org

Editor comments:

Thanks for your revisions. As you see, one reviewer is satisfied, but the other one and our AE are not yet. Please be aware that if in your revision you do not address all queries successfully we will not be able to continue the manuscript further, because it would be the second resubmission. Best success in your revisions.

Associate Editor Comments to Author (Dr Cynthia Downs):

Thank you for resubmitting “Avian malaria-mediated population decline of a widespread iconic bird species.” The revised manuscript was reviewed by two of the reviewers who reviewed the original submission and me. In general, the reviewers were satisfied with the revisions that were made to the manuscript. Although the reviewer that raised concerns about the statistics included in the original manuscript did not review the revisions, I am satisfied with the explanation as to why two statistical frameworks were used to analyze their data.

Reviewer two raised concerns that the introduction doesn't provide enough context for the study, specifically with regards to investigations about the decline of house sparrow population in the United Kingdom. Please expand the introduction. Please also address reviewer two's comments about the uses of the terms parasite infection, parasitaemia, and infection intensity and address their line comments about the methods detailed in the PDF they provide.

It appears that you collected data of abundance for each population (methods presented in 79-93), but these data are not presented in the results. These measures would provide an index of your sampling effort and could help strengthen the conclusions drawn from the results. Please include these data as a table or figure in an electronic supplement and discuss them in relation to the other results presented.

Finally, as noted by reviewer two, Table S2 appears to be missing from the supplementary material. Please rectify this in your next submission or explain why it has not been included.

I greatly enjoyed reading this revision.
Best regards,
Cynthia Downs

Reviewer comments to Author:

Reviewer: 2

Comments to the Author(s)

Thank you for taking the time to thoughtfully address and respond to reviewer comments and suggestions. Looking forward to seeing this manuscript published.

Reviewer: 1

Comments to the Author(s)

The manuscript was significantly improved from its previous version. However, I think the Introduction is still too short and lacking of information. In my opinion, some parts from the large Discussion might be modified and moved to the Introduction, as for example the last paragraph of the Discussion (L408-432).

The terminology use is still not appropriate on some places (L58, L62). Please be aware that "parasitaemia " and "infection intensity" have the same meaning in parasitology. I would recommend to stay consistent and use one of the terms. For blood parasites "parasitaemia" is more commonly used.

Why I could not find supplementary Table 2? Is it wrongly entitled as TableS 3?

Author's Response to Decision Letter for (RSOS-182197.R1)

See Appendix B.

Decision letter (RSOS-182197.R2)

13-Jun-2019

Dear Dr Dadam,

I am pleased to inform you that your manuscript entitled "Avian malaria-mediated population decline of a widespread iconic bird species" is now accepted for publication in Royal Society Open Science.

on behalf of Dr Cynthia Downs (Associate Editor) and Kevin Padian (Subject Editor)
openscience@royalsociety.org

Associate Editor Comments to Author (Dr Cynthia Downs):

Thank you for addressing my comments and the comments from the reviewers. This study provides an important link between host survival with the intensity of their parasite infections.

Appendix A

Response to the Associate Editor.

- I suggest extending the introduction to provide more context for how the current study fits into the literature as suggested by two of the reviewers.

We have extended the Introduction, as suggested.

- I also recommend addressing the comments made by reviewer 3 about the statistical modeling approach. Particularly with regards to clarifying the statistical methods used and reviewer 3's suggestion to apply an information-theoretic approach to analyse population growth.

We have replied to the reviewer's comment below.

- In addition, the models used to develop the model-averaged best model for survival included the model set within 2 delta AICs of the top model. It is more common to use the model set within 4 delta AICs of the top model. Please explore this more conservative approach to ensure that it does not change the interpretation of the data.

We have considered models within 4 delta AICs. The results and their interpretation have not changed.

- Also, please address the suggestions from the reviewers about clarifying methods that are important for interpreting the scientific soundness of the study presented.

We have addressed those comments, more details below.

Response to Reviewers.

Reviewer: 1

- The reviewed manuscript presents interesting and novel findings on the influence of pathogens such as protozoan parasites (Haemosporida and Coccidia, Eucoccidiorida), bacteria, and helminths on populations declines of house sparrows (*Passer domesticus*) in Lindon, UK. The paper is well written and well structured. However, I think that section "1. Introduction" have to be expanded and more informative. Perhaps the following studies worth citing and discussion:

We thank the reviewer for these positive comments and suggestions.

- Loiseau, C., Zoorob, R., Robert, A., Chastel, O., Julliard, R. & Sorci, G. (2011) *Plasmodium relictum* infection and MHC diversity in the house sparrow (*Passer domesticus*). *Proceedings. Biological sciences / The Royal Society* 278, 1264–1272.

Added in the Discussion (Line 404).

- Bichet, C., Sorci, G., Robert, A., Julliard, R., Lendvai, Á.Z., Chastel, O., Garnier, S. & Loiseau, C. (2014) Epidemiology of *Plasmodium relictum* infection in the house sparrow. *Journal of Parasitology* 100, 59–65.

Added in the Discussion (Line 320).

- Marzal, A., Ricklefs, R.E., Valkiūnas, G., Albayrak, T., Arriero, E., Bonneaud, C., Cziráj, G. a, Ewen, J., Hellgren, O., Hořáková, D., Iezhova, T.A., Jensen, H., Križanauskienė, A., Lima, M.R., de Lope, F., Magnussen, E., Martin, L.B., Møller, A.P., Palinauskas, V., Pap, P.L., Pérez-Tris, J., Sehgal, R.N.M., Soler, M., Szöllosi, E., Westerdahl, H., Zetindjiev, P. & Bensch, S. (2011) Diversity, loss, and gain of malaria parasites in a globally invasive bird. PLoS one 6, e21905.

Whilst a very interesting reference, we did not add it because we found it difficult to contextualise it. It focuses on house sparrows as an invasive species and their having fewer haemoparasites than in their native range, a topic that was not covered in our manuscript.

- For me was also quite difficult to move between methodology in the section “2. Methods” and supplementary materials S1-S4 in order to get familiar with it. I would recommend if possible gathering them together.

We have combined the text in S1-S4 with the main Methodology.

- Another questionable part of the methodology is how *Atoxoplasma* and *Isospora* species was differentiated despite the discussion on L210-216. Since *Atoxoplasma* was suggested as junior synonym of *Isospora*, how they were separated based on the blood smears or fecal samples?

We have moved the explanation of how *Atoxoplasma* and *Isospora* were identified from the Discussion to the Methodology and we have added extra clarification (Lines 158-162). Although considered to be the same parasite, the two names are used to differentiate the intestinal phase (*Isospora*) and the blood phase (*Atoxoplasma*),

- The scientific terminology was not always appropriately used. For example L 161 “intensity of parasitism” Perhaps you mean intensity of infection or parasitaemia? For blood parasites “parasitaemia” is more commonly used. I would recommend the following paper to solve similar issues:

Bush, A.O., Lafferty, K.D., Lotz, J.M. & Shostak, A.W. (1997) Parasitology meets ecology on its own terms: Macrogolis et al. Revisited. *Journal of Parasitology* 83, 575–583.

We have changed “parasitism” for “parasitaemia” in most case, aside from when we considered “intensity of infection” to be more appropriate.

- Line 268 “inoculation dose”, please note that infection initiated via artificial inoculation with infected blood and via vector bite are different techniques and it is not correct to compare them equally! On L 272 you have multiple inoculations with sporozoites of *P. relictum*, which is not the same as studies 45 and 46.

We agree and we have modified the text accordingly (Lines 375-379).

- In my opinion it worth discussing the abundance of vectors of *P. relictum*, *Culex pipiens* if you have such data for the different sparrow populations in your study rather than compare with experimental infections initiated with inoculation of infected blood.

Unfortunately we do not have the data but we agree that it would be very interesting to look at *C. pipiens* abundance across the different house sparrow populations. We have added this point to the Discussion (Line 369).

Reviewer: 2

Abstract:

A couple wording suggestions:

- Line 22: "Populations of house sparrow (*Passer domesticus*) have declined in many European towns and cities, but the causes of these declines remain unclear."

Text modified accordingly (Lines 21-23).

- Line 26: "*Plasmodium relictum* infection was found at higher prevalences (averaging 74%) in suburban London house sparrows than previously recorded in any wild bird population in Northern Europe."

Text modified accordingly (Lines 25-27).

- Line 31: "Our findings indicate that *P. relictum* infection may be influencing house sparrow population dynamics in suburban areas."

Text modified accordingly (Lines 31-32).

Introduction:

- Starts well with broad themes, but the focus needs to narrow down to the present study. The third sentence of the first paragraph (Line 42) is vague, what are the potential demographic impacts of parasites on birds?

We have clarified this point (Line 41-45) and added more explicit examples (Lines 46-50).

- Also, avian malaria is in the paper's title, but there is no mention of it in the Introduction. Introduce *Plasmodium* perhaps around Line 52 when mentioning parasites are generally understudied as a potential cause of avian population declines.

It is now mentioned more explicitly amongst the examples of potential demographic impact on birds (Line 46).

Wording:

- Line 45: "The house sparrow (*Passer domesticus*, hereafter sparrow) used to be one of the most common species of bird in the United Kingdom (UK) ..."

Text modified accordingly (Lines 52-53). We have also changed "common" with "abundant".

Methods

- Line 64: States the study was conducted between November 2006 and September 2009 but section (b) of Methods says surveys of male sparrows were completed between 2005 and 2009. Does it matter for this study that annual surveys of male sparrows were conducted a year before this study began?

Yes, that is correct. The annual surveys measured change in house sparrow abundance between 2005 and 2009, and the parallel parasitology work was conducted between 2006 and 2009. It was useful to have as many years as possible contributing to our measure of colony growth rate.

- Line 67: Is 4 km a significant distance for house sparrow populations or is this just a minimum distance between study populations?

This was the minimum distance between neighbouring colonies and it is useful to quote to demonstrate that our study colonies were probably demographically isolated from each other. Natal and breeding dispersal rarely exceed 2km in this species. We have modified the text to clarify this point (Lines 75-76).

- Line 87: Was data from the "pilot study" used in the analysis?

No, it wasn't, as it had taken place only at one site

- Line 90: Write out full years: 2007/2008 and 2008/2009

Text modified accordingly (Lines 113-114).

- Lines 96-97: You state you identified the relative abundance of parasites in each faecal sample (but not swab). What are the swabs you are referring to here?

We are referring to cloacal swabs, which were used only for bacteriology as their tip, which had faecal matter, had been preserved in charcoal medium and hence was not suitable for parasitology. We were clarifying that swabs had not been used for parasitology. We have combined the supplementary material Methodology with the main text and we have added a short clarification within the parenthesis (Lines 125-126).

- Line 125: Wording – "Separate analyses were conducted first based on infection rates of all birds, and then of adults and juveniles separately. Too few individuals were sexed to allow sex-specific analyses.

Text modified accordingly (Lines 188-192). We have inserted an additional sentence splitting the two.

- Line 137: Add “n =” in front of 117 when reporting sample size

Text modified accordingly (Line 202).

- Lines 166-169: Run-on sentence. Each statement here should be in a single sentence.

Text modified accordingly (Lines 257-262).

- Section (c): Report sample size with all statistics

Text modified accordingly (Lines 265-279).

Results:

- Line 176: You report there was no association between prevalence of Plasmodium and local sparrow population growth in the Discussion but this should also be reported here in the Results.

Text modified accordingly (Lines 267-269).

- Line 228: You switch from “juveniles and adults” to “adults and first years”. Stay consistent.

Text modified accordingly. We use “juvenile” throughout.

- Line 235: Plasmodium lineages cannot be “globally widespread” and “endemic to the UK”.

Perhaps use “native” instead of “endemic” here?

Text modified accordingly (Line 337).

Discussion:

- In the Introduction you mentioned the multiple hypotheses put forth to explain house sparrow decline. It might be informative to discuss those studies in the Discussion, especially if some of those factors may be contributing to the overall decline along with avian malaria.

Text modified accordingly (Lines 408-432).

- Line 245: Bichet et al.?

Changed as suggested (Line 347)

- Line 269: Are you referring to house sparrows or more broadly to sparrows as a family? Either way, specify here.

We refer to house sparrows. In the Introduction (Line 52) we have specified that we would call them “sparrow” from then on.

- Line 270: Full species name is European starling.

Text modified accordingly (Line 374).

- Line 271: Atkinson et al.?

Changed as suggested (Line 377)

- Line 277: This paragraph does not fit well here without prior introduction. This information either needs to be presented in the Methods and Results sections, or not at all. It might be more appropriate as a stand-alone publication such as a “Short Communication”.

We have now moved text from the Supplementary Material to introduce testing for an effect of air pollution in the Methods (Lines 214-233) and Results (Lines 294-299) sections. This now flows through Methods to Results and Discussion.

- Line 296: Wording – “Inbreeding coefficient has been associated with a high level of homozygosity in sparrows [50], therefore individuals with lower heterozygosity may be less able to mount an immune response [51].

Text modified accordingly (Lines 404-406)

Tables and Figures:

- Figure 1: Figures should be able to stand-alone, title needs more description – “Map of house sparrow (*Passer domesticus*) study sites in London, indicating whether or not counts of territorial male house sparrows showed a declining trend over the period 2005-2009”.

Text modified accordingly

- Figure 2: Figure should stand-alone. Report that parasite prevalence and intensity was measured in house sparrows (*Passer domesticus*). Report what the line represents in figure a) in the description. Also report that each symbol represents a distinct house sparrow population. This might read better as two separate figures.

The Figure heading has been changed accordingly. The two graphs have been kept as part of the same figure as we feel it renders the contrast between prevalence and intensity easier to visualise.

- Figure 3: "Relationship between intensity of *Plasmodium relictum* infection and monthly survival of (a) juvenile and (b) adult house sparrows (*Passer domesticus*), illustrating the age*plasmodium survival model with independent slopes (Model 2 Table S4). Ticks on the x-axis indicate infection intensity per individual sparrow."

Figure heading changed as suggested.

- Table 1: Use the percent sign when reporting each percent. You've placed the percent sign in the column titles but I think it would be clearer to write it out next to each percentage.

Percentage sign added.

Reviewer: 3

Comments to the Author(s)

Major comments:

- L. 28-30 – That other parasites were less prevalent and no relationship with house sparrow survival, etc. assumes that co-infection is always 'negative' where some (limited) literature suggests that the negative effects of co-infection do not worsen additively with each new parasite infection. Consider addressing this briefly in the Discussion paragraph starting at L. 244.

Consideration of this has been added to the Discussion with examples provided (Line 354)

-Analyses – I appreciate the care that was taken in investigating the effects of parasitism on house sparrows. However, several issues emerged that should be considered to make the analytical approach and inferences derived from it stronger. These include addressing the following:

37) A) L.122-123 – How were the analyses (models) weighted by prevalence or intensity? This was confusing to me.

We have clarified this point in the text (Lines 187-188). Weighting of observations within GLMs like this is a fairly standard statistical technique.

- B) L. 130 – What was the CJS 'extension' that was used? (May be simply "CJS model")

Yes, it was just CJS model, and we have modified the text accordingly (Line 195).

- C) CJS modeling approach – If I understood correctly, the mark-recapture analyses had two phases. One was to ask about the ecological factors associated with apparent survival and detection. Why take this stepped approach? Consider providing additional detail and reasoning for this approach directly in the MS. Related, as I understand it, the infection intensity values were those determined using the blood sample taken when the birds were first caught and banded. Couldn't infection intensity change over the course of the study? It IS interesting that survival varies with infection intensity and age but are infection intensity

and age correlated? If so, the inferences about age and intensity get complicated. Also, what about the simple relationship between infection status at first capture and apparent survival? This seems like an important question that is not addressed but could be easily.

We have taken a two-step approach (model simplification followed by testing for parasite effects) in order to maximise statistical power at the second stage. This is a standard approach in such survival analyses. We have modified the text to clarify our approach (Lines 206-207).

We agree that infection intensity could change over the course of the study and we would have liked to test this, but too few sparrows were captured to allow such an analysis. Mean intensity of infection was higher in juveniles (mean Plasmodium= 24) than adults (mean Plasmodium=12), but we show a similar negative relationship existed between survival and plasmodium intensity for both age classes (Fig 3).

We feel the relationship between infectious status (infected/not infected) and apparent survival would have less resolution than our current approach and probably be difficult to interpret. This is because in a category “infected”, for example, we would combine house sparrows that had very low intensity with those individuals with high intensity. Should the result show that simply being infected does not significantly reduce chances of survival, it would have to be interpreted as a spurious result because it would go against the more detailed analyses that take intensity of infection into consideration and which show that at high levels intensity of infection does decrease survival probability. If, on the other hand, results showed that being infected is enough to have lower chances of survival, it would give a less-comprehensive picture than the current intensity-based survival analysis provides.

- D) Mixing paradigms – Avoid mixing the information-theoretic (IT) approaches used to report the capture-recapture model selection results with null hypothesis testing that is used for the population growth analyses. Consider reworking the population growth analyses in an IT framework; this allows discussion of the weight of evidence for competing models and lets the factors of interest ‘duke it out’ with respect to which one or ones best explain the variation in the data. This approach also allows a deeper analysis of the combinations of factors – including infection intensity - that might be associated with changes in population size (is it size or is it growth like lambda?) A minor point – significance wording is not needed when reporting effect sizes from the ‘best model’ under an IT framework—that veers into mixing paradigms within the same analysis instead of just within the same paper.

Ideally we would have preferred to avoid mixing paradigms by applying an IT framework to the population growth analyses as suggested, but it was technically impossible to do this because sample size differed between models (AIC is only comparable across models when the underlying data structure remains constant but we lacked measures of several age-specific parasite metrics at some colonies). We recognise that this was not clear in the manuscript and we have tried to clarify this point (Lines 190-191). In particular, prevalence and intensity of infection in juveniles are modelled on two fewer sites than the other models, because no juveniles were successfully sampled at those sites. This prevented us using an IT framework on population growth analyses, since the juvenile models are pivotal to the argument put forward in this manuscript.

We then considered dropping the IT framework from the survival analyses in order to avoid mixing paradigms between different analyses. However, we could not see a defensible way of conducting our survival analysis without using an AIC-based approach. For example, how would we have identified a base model against which to assess the effects of parasite status?. We therefore retained the AIC approach in the survival analyses, which is standard practice for studies of this type. We have been careful not to mix paradigms within any given analysis and, whilst we recognise that not mixing paradigms within a single MS would be ideal, there are many contemporary studies (including in Open Science) that are forced to do this for different analyses within a single paper.

- The NO₂ analyses could also be done in an IT framework. Consider consulting Anderson 2007 - Model Based Inference in the Life Sciences: A Primer on Evidence and associated material for more on analyses in an IT framework.

The same technical problem to that described above applies here too: due to differences in sample size we could not adopt an IT framework for these analyses.

Minor edits to consider:

- L. 30 – Add “was shown” after “abundance”

The sentence has been modified and split into two for clarity (Lines 28-30)

- L. 50 – Add “negative” before “impacts of pollutants”

Text modified accordingly (Line 57).

- L.57 – 73% here is different than the value cited in the Abstract (72%)

Text has been updated with the latest figures (Line 64).

- L. 68 – consider replacing “travel” with “disperse”; when I first read it I thought to myself “They travel a lot through flight...maybe kilometers in a day”

We have specified that we were talking about natal and breeding dispersal to clarify that we are referring to adult movements (Lines 74-76).

- L.189 – “AIC support” would be better stated as “Akaike weight” or “AICc weight” (the AICc is the small sample size corrected form of AIC and is often used almost by default because $AIC = AICc$ when sample sizes are large)

Text modified accordingly (Line 283).

- L. 250 – correct spelling to “exception”

Text modified accordingly (Line 352)

- L. 307 – consider replacing “reduced” with “decreased”

Text modified accordingly (Line 441).

Appendix B

Response to the Associate Editor.

- It appears that you collected data of abundance for each population (methods presented in 79-93), but these data are not presented in the results. These measures would provide an index of your sampling effort and could help strengthen the conclusions drawn from the results. Please include these data as a table or figure in an electronic supplement and discuss them in relation to the other results presented.

Thank you for your comment. The abundance data were already presented as an integral part of the results (Results sections (c) and (e), Fig. 2 and Table 3), which show that cross-colony variation in growth rates through time is correlated to *Plasmodium* intensity of infection (Figure 2b). We agree, though, that it may assist the reader to have access to the raw data from the house sparrow survey, so have added it as a supplementary table, now Table S1. The proportion of birds sampled in each colony compared to the colony count can be inferred from comparing Table S1 and Table S3, but we did not feel it warrant a specific mention in the main manuscript.

Response to Reviewers.

Reviewer 1

-I think the Introduction is still too short and lacking of information. In my opinion, some parts from the large Discussion might be modified and moved to the Introduction, as for example the last paragraph of the Discussion (L408-432).

Thank you for your suggestion, we have moved the last paragraph of the Discussion to the Introduction (Lines 57-79).

- Please be aware that “parasitaemia” and “infection intensity” have the same meaning in parasitology. I would recommend to stay consistent and use one of the terms. Furthermore, for me is not clear in the sentence what you mean under “parasitaemia” and “infection intensity” (L62). [...]

Thank you for your comment. Parasitaemia simply means the presence of parasites in the blood (e.g. Gabra, M.S., Grossiord, D., Perrin, L.H., Shaw, A., Cheung, A. and McGregor, I.A., 1986. Defined *Plasmodium falciparum* antigens in malaria serology. *Bulletin of the World Health Organization*, 64(6), p.891) and this is the sense in which we use the term. We recognise, however, that it is increasingly used to mean parasite ‘abundance’ or ‘intensity’ of infection in clinical journals, therefore we have changed “parasitaemia” to “infection intensity” (or “intensity of infection”) throughout the manuscript except where the term “parasitism” better depicted the required meaning.

We have also modified the last sentence of the Introduction, as suggested, to make it clearer that we are referring to parasite infection (“parasitism”) and subsequently to intensity of infection. (Lines 81-82)

- How long the fixated smears were stored before staining? (Line 147)

They were stained the next day in most cases but at peak times some slides were stained a few days later. The text has been amended to state that slides were stained within a week. (Line 166)

- Do you mean tested positive for Plasmodium sp, based on blood smears? How these 29 individuals were selected? Did you use equal samples from each study site, host age or sex? (Line 165)

The 29 samples were selected from positive ones as seen from blood smears, now clarified in the text (Lines 185-186). The 29 individuals were selected to represent all sites in roughly equal numbers (text modified accordingly, Line 185), although they were arbitrarily chosen among those available for each site and from which we had a large enough blood sample. Also, we ensured that we included adults and juveniles of both sexes).

- Was the overall prevalence estimated based on blood smears only? (Line 251)

Yes, it was based on microscopy, we have now specified this in the text (Lines 271-272).

- Plasmodium in italic (Line 322 and 354)

Thank you, they have been rectified (Lines 342 and 374).

- I think this paragraph (Lines 408-432) can be easily moved in the Introduction section to give better background on the main aim of the study.

Thank you for your suggestion, we have moved the paragraph you have suggested to the Introduction (Lines 57-79).

- Why I could not find supplementary Table 2? Is it wrongly entitled as Tables 3?

Our apologies, indeed it was mis-labelled and it was the file erroneously named Table S3. We have now added a table in the supplementary material, therefore the table is now correctly named Table S3.

Reviewer 2

Thank you for taking the time to thoughtfully address and respond to reviewer comments and suggestions. Looking forward to seeing this manuscript published.

Thank you very much for your input.